# Pharmacological inhibition of catalase induces peroxisome leakage and suppression of LPS induced inflammatory response in Raw 264.7 cell

**Yizhu Mu**[�], **Yunash Maharjan**[�], **Raghbendra Kumar Dutta**[�], **Xiaofan Wei, Jin Hwi Kim**[ID], **Jinbae Son, Channy Park, Raekil Park**[ID]*

Department of Biomedical Science & Engineering, Gwangju Institute of Science & Technology, Gwangju, Republic of Korea

y These authors contributed equally to this work.
* rkpark@gist.ac.kr

## Abstract

Peroxisomes are metabolically active organelles which are known to exert anti-inflammatory effects especially associated with the synthesis of mediators of inflammation resolution. However, the role of catalase and effects of peroxisome derived reactive oxygen species (ROS) caused by lipid peroxidation through 4-hydroxy-2-nonenal (4-HNE) on lipopolysaccharide (LPS) mediated inflammatory pathway are largely unknown. Here, we show that inhibition of catalase by 3-aminotriazole (3-AT) results in the generation of peroxisomal ROS, which contribute to leaky peroxisomes in RAW264.7 cells. Leaky peroxisomes cause the release of matrix proteins to the cytosol, which are degraded by ubiquitin proteasome system. Furthermore, 3-AT promotes the formation of 4HNE-IκBα adduct which directly interferes with LPS induced NF-κB activation. Even though, a selective degradation of peroxisome matrix proteins and formation of 4HNE- IκBα adduct are not directly related with each other, both of them are could be the consequences of lipid peroxidation occurring at the peroxisome membrane.

## Introduction

Peroxisomes are essential organelle widely known for its role in the oxidation of very-long chain and branched chain fatty acids as well as biosynthesis of bile acid and ether phospholipid [1, 2]. Many of the oxidation reactions carried out by peroxisome resident oxidases produce $H_2O_2$ as a by-product, which is efficiently converted to water and oxygen by peroxisome localized catalase and other organelle peroxidases to maintain cellular redox balance [3, 4]. Numerous independent studies have validated that peroxisome deficiency and condition in which cell loss its' ability to maintain catalase function lead to an increase of oxidative stress [5–8]. Moreover, oxidative stress and inflammation are intimately connected to pathophysiological events in several chronic diseases [9]. A possible regulatory role of peroxisomes in the inflammatory pathway was speculated from the several functional attributes including its role in (1) redox homeostasis, (2) biosynthesis of docosahexaenoic acid (DHA) and eicosapentaenoic acid

**Data Availability Statement:** All relevant data are within the paper and its Supporting information files.

**Funding:** This work was supported by: •National Research Foundation of Korea (NRF) under grants funded by the Korean government No. 2019R1A2C208608012 •"GIST Research Institute (GRI) IIBR" grant funded by Gwangju Institute of Science and Technology (GIST) in 2020.

**Competing interests:** The authors have declared that no competing interests exist.

which are the precursors for the mediators of inflammation resolution, and (3) its' pivotal role in metabolism of inflammatory mediators such as leukotrienes and prostaglandins [10–12]. Recently, a direct anti-inflammatory role of peroxisome abundance in LPS mediated inflammatory pathway has been reported [10].

3-Aminotriazole (3-AT) is a well-known irreversible catalase inhibitor [13, 14]. It has been employed in the numerous studies to analyse the effect of ROS on cytotoxicity and adipogenesis [14, 15]. Previously, we have demonstrated that 3-AT attenuates cobalt chloride induced cytotoxicity through inhibiting of ROS generation and pro-inflammatory cytokines in a catalase independent manner [16, 17]. Furthermore, we have recently reported that catalase inhibition by 3-AT induces pexophagy through accumulation of ROS [18]. Therefore, it is reasonable to speculate that 3-AT might modulate LPS induce pro-inflammatory pathway in macrophages either through inhibition of catalase or enhancement of pexophagy.

In this study, we explore a possible immunomodulatory role of 3-AT on LPS induced proinflammatory response in RAW264.7 cells. We found that 3-AT specifically induces peroxisomal ROS generation which causes peroxisome leakage and promotes proteasomal degradation of peroxisome matrix proteins with no relevance to pexophagy. Furthermore, 3-AT prevents phosphorylation of IκBα through promoting the formation of 4HNE-IκBα adduct which directly interferes with LPS induced NF-κB activation and expression of pro-inflammatory cytokines.

## Materials and methods

### Reagent and antibodies

3-Amino-1, 2, 4-triazole (#A8056), N-acetyl-l-cysteine (#A7250) and Chloroquine (#C6628), were purchased from Sigma-Aldrich. Recombinant mouse TNF-α (#MTA00B) and IL-6 (#M6000B) enzyme-linked immunosorbent assay (ELISA) kits (Quantikine) for cytokine analysis were bought from R&D Systems (Minneapolis, MN). Anti-PMP70 (#sab4200181, Sigma-Aldrich), anti-Catalase (#ab16731, Abcam), anti-SQSTM1/p62 (#H00008878-M01, Abnova), anti-LC3 (#L8918, Sigma-Aldrich), anti-Pex5 (#GTX109798, GeneTex), anti-Pex7 (#20614-1-AP, Proteintech), anti-DBP (#TA308904, OriGene), anti-ACOX1 (#10957-1-AP, Proteintech), anti-UBXD8 (#NB100-1296, Novs), anti-HA (#ab130275, Abcam), anti-Pex1 (#13669-1-AP, Proteintech), anti-Pex16 (#14186-1-AP, Proteintech), nti-Pex3 (#247042, Abcam), anti-Pex9 (PA5-22129, Invitrogen), anti-Tomm20 (#ab56783, Abcam), anti-NF-κB (#sc-8008, Santa Cruz), anti-IκBα (#9242S, CST), anti-p-IκBα (#2859S, CST), anti-4-HNE (#ab46545, Abcam) and anti-β-actin (#sc-47778, Santa Cruz).

### Cell culture

RAW 264.7 cells were cultured in high-glucose Dulbecco's modified Eagle medium (DMEM, Gibco-BRL, Grand Island, NY, USA) supplemented with 10% fetal bovine serum (FBS, Gibco-BRL), 100 IU/ml penicillin (Invitrogen), and 100 μg/ml streptomycin (Invitrogen) at 37˚C and 5% $CO_2$ in a humidified atmosphere.

### Plasmid transfection

Cells were transiently transfected with Hyper-SKL (gifted from Dr. Dong-Hyung Cho, Kyung Hee University, South Korea) [19] and HA-ubiquitin plasmids (#18712, addgene) using Lipofectamine 3000 (#L3000015, Invitrogen) according to the manufacturer's instruction.

## RNA isolation and real-time qPCR analysis

Total RNA was extracted from the cells using TRIzol reagent (#15596018, Life Technologies). A reverse transcription kit (#04379012001, Roche) was used to transcribe cDNA, and then qPCR was performed with cDNA as a template using a Light Cycler system with FastStart DNA Master SYBR Green (#06402712001, Roche). The mouse primer sequences (forward and reverse, respectively) were as follows: *Catalase* (5′-ccttcaagttggttaatgcaga-3′ and 5′-caagttttttgatgccctggt-3′), *PMP70* (5′-aagcagacaatccactcagtctt-3′ and 5′-cccatagaaaaccgaaagaaaa-3′), *ACOX1* (5′-cgccagtctgaaatcaagaga-3′ and 5′-gctgcgtctgaaaatccaa-3′), *DBP* (5′-gggctgtcattcaactttgc-3′ and 5′-ggaagtggcttatacagctcca-3′), *TNF-α* (5′-ctacctccaccatgccaagt-3′ and 5′-gcagtagctgcgctgataga-3′), *IL-6* (5′-tcgtggaaatgagaaaagagttg-3′ and 5′-agtgcatcatcgttgttcataca-3′), *IL-1β* (5′-agttgacggaccccaaaag-3′) and 5′-agctggatgctctcatcagg-3′) and intracellular control *36B4*, (5□-cactggtctag-gacccgagaag-3□, 5□-ggtgcctctggagattttcg-3□).

## Subcellular fractionation

For subcellular fractionation, the harvested cell were suspended in in Buffer A (50 mM HEPES pH 7.6, 1.5 mM MgCl2, 10 mM KCl) with inhibitors of protease (# P3100-001, GenDEPOT) and phosphatase (#P3200-001, GenDEPOT). Cell lysis was achieved by passing cells through a 22-G1 needle 30 times and were centrifuged at 1,000 g for 5 min at 4˚C (C1). Pellet obtained from C1 was suspended in Buffer C (20 mM HEPES pH 7.6, 1.5 mM MgCl2, 0.42 mM NaCl, 2.5% glycerol) with inhibitors of protease and phosphatase, rotated for 1 h at 4˚C and centrifuged at 18,500 g for 30 min at 4˚C. Supernatant thus obtained comprised the nuclear fraction. Supernatant obtained from C1 was centrifuged at 18,500 g for 30 min at 4˚C (C2). The supernatant obtained from C2 represented cytosol fractionation whereas the pellet represented as the membrane fraction. Lysis buffer [10 mM Tris–HCl pH 6.8, 100 mM NaCl, 1% sodium dodecyl sulfate (SDS)] with protease and phosphatase inhibitors were added to membrane fractionation and incubated for 30 min at room temperature. SDS loading buffer (60 mM Tris–HCl pH 6.8, 2% SDS, 1% b-mercaptoethanol, 10% glycerol, and 0.02% bromophenol blue) was added to all subcellular fractionation and denatured at 95˚C for 5 min. The subcellular fractionation obtained were used for immunoblot analysis.

## Immunofluorescence

Cells grown on coverslips were washed with phosphate buffered saline (PBS, pH 7.4), fixed with 4% paraformaldehyde for 15 min at room temperature. Cells were then rinsed PBS, permeabilized with 0.25% Triton X-100 for 5 min, followed by blocking with 3% bovine serum albumin for 1 h at room temperature. Cells were then incubated with primary antibodies in 3% bovine serum albumin at 4˚C for overnight, rinsed with PBS, and labeled with fluorescent Alexa Fluor 488 or Alexa Fluor 568 (molecular probes)-conjugated secondary antibodies (1:500) in dark for 1 h at room temperature. To detect the nuclei, coverslips were mounted on slides with the Prolong Gold antifade reagent containing DAPI (4'5-diamidino-2-phenylindole; #P36931, Invitrogen) and examined under either Olympus Fluoview 1000 confocal laser scanning microscope or fluorescence microscope (IX71, Olympus.

## Measurement of ROS

ROS level was measured with the dye 2′,7′-dichlorofluorescein diacetate (DCFH-DA) as previously described [20] with slight modification. Raw 264.7 cells transfected with Hyper-SKL

plasmid were cultured in the presence or absence of 3-AT. DCFH-DA (1 μM) was added to cells and, followed by incubation for 45 min at 37˚C and 5% $CO_2$. Cells were then washed with PBS.

To detect the nuclei, coverslips were mounted on slides with the Prolong Gold antifade reagent containing DAPI and observed under fluorescence microscope.

DCFH-DA intensity was measured from full image with similar cell distribution in each condition by Image J software. Intensity from at least 10 full images was measured per condition for each experiment. The results were interpreted as arbitrary unit or fold of control.

## Measurement of HyPer SKL intensity

Area around the HyPer-SKL fluorescent signal was considered as region of interest for the measurement of intensity by Image J software. Intensity from at least 30 HyPer-SKL transfected cells was measured per condition for each experiment. The results were interpreted as fold of control.

## Catalase activity assay

Catalase activity was determined using Catalase Potassium Periodate Kit according to r manufacturer's instruction ((#707018, Cayman).

## Immunoprecipitation

Raw 264.7 cell pellet was homogenized in 0.5 ml Seize 2 buffer (150 mM NaCl [pH 7.2], 50 mM Tris HCl) containing 0.2% NP-40 (Calbiochem, San Diego, CA, USA) mixed with $1 \times$ protease and phosphatase inhibitors cocktail. Approximately 600 μg of cell lysate was subjected to immunoprecipitation with anti-HA or anti-IκBα bound to Protein A/G beads. Total volume of reaction mixture was maintained to 300 μl. Immunoprecipitation reaction was carried by rotating the reaction mixture at 4˚C for 8 h, followed by centrifugation at 14000 RPM for 5 min, and supernatant was collected. Laemmli buffer (4% SDS, 20% glycerol, 10% 2-mercaptoethanol 0.004% bromophenol blue, 0.125 M Tris HCl) pH 6.8 was then added to supernatant so as to adjust the final concentration of SDS to 1X and boiled at 97˚C for 10 min. Protein A/G beads were washed three times with Seize 2 buffer and 300 μl of 1X Laemmli buffer was added and boiled at 97˚C for 10 min and it was considered as pellet fraction. Equal volume of both pellet and supernatant was subjected to immunoblot analysis with necessary antibodies.

## Proteinase K protection assay

Cell pellets were suspended in a homogenization buffer (50 mM Tris HCl, 1 mM EDTA, 250 mM sucrose, pH 7.2) with $1 \times$ protease and phosphatase inhibitors cocktail by passing through 23 G needle for 8 times followed by centrifugation at $1000 \times g$ for 5 min at 4˚C. The resulting post nuclear supernatants were separated into three groups: group1, 1% Triton X-100; Group2, Proteinase K (Roche) 0.1 μg/ml; Group 3, combination of Proteinase K 0.1 μg/ml with 1% Triton X-100. All groups were incubated on ice for 30 min. Reaction was stopped by the addition of 1 mM PMSF and samples were processed for Western blot assay.

## Whole cell lysis

Cells were harvested by scrapping from culture dish and were centrifuged at 1000 g for 5 min at 4˚C. The pellets obtained were lysed on ice cold RIPA buffer (20 mM HEPES pH 7.5, 150 mM NaCl, 1% Triton X-100, 1% sodium deoxycholate, 1 mM EDTA) mixed with protease and phosphatase inhibitors and centrifuged at 18,500 g for 10 min at 4˚C. SDS loading buffer was

added to the supernatant and denatured at 95°C for 5 min. The supernatant were used for immunoblot analysis.

The level of protein expression was quantified using Image J Software. β-actin was used as protein loading controls.

### Statistical analysis

Data obtained from at least three independent experiments were shown as ± SD. Two-tailed Student's t-test was employed for comparison of the significant differences between two groups.

## Result

### Pharmacological inhibition of catalase caused by 3-AT selectively promotes the degradation of peroxisomal matrix proteins in RAW264.7 cells

RAW264.7 cells were treated with different concentrations of 3-AT to measure the enzymatic activity of catalase. As shown in (Fig 1A), treatment with 3-AT decreased the catalase activity in a dose dependent manner. Cell viability was not affected by all the tested concentrations of 3-AT at different time intervals (S1A–S1D Fig). We investigated if 3-AT alters the expression of proteins related to peroxisome abundance and function, autophagy and mitochondrial

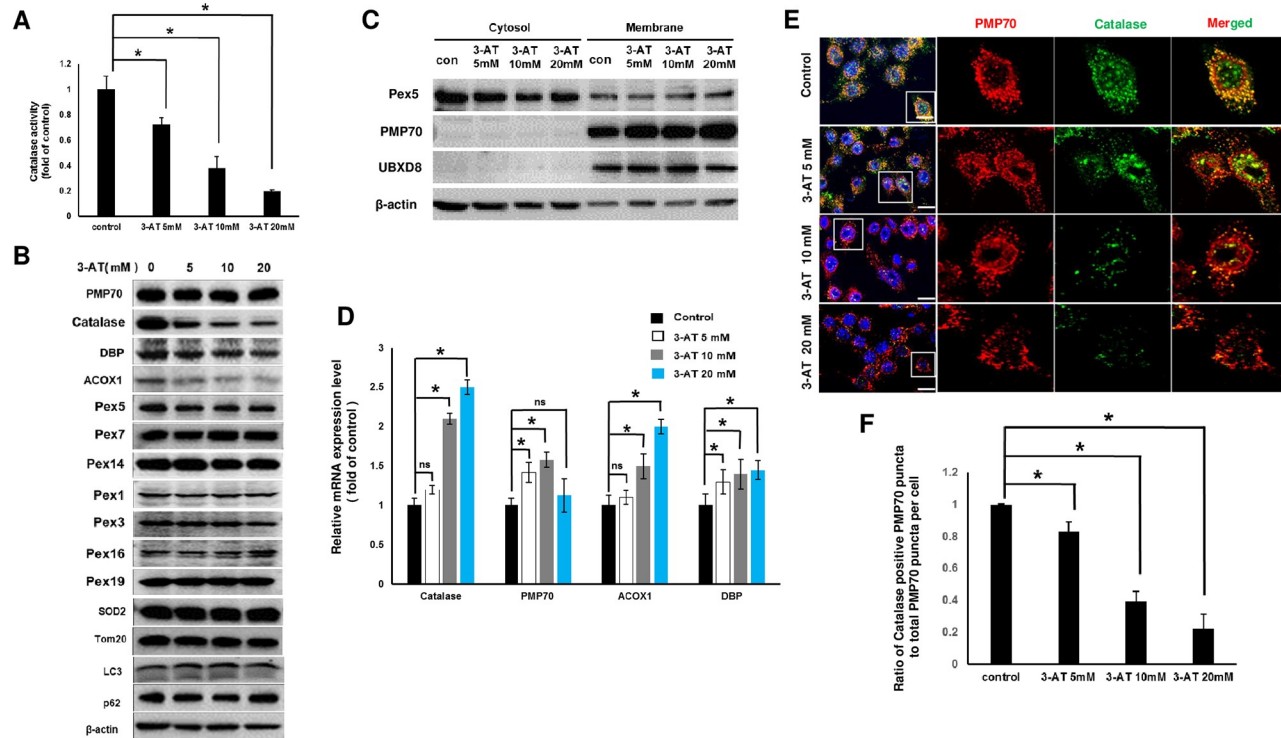

**Fig 1. 3-AT selectively decreased peroxisomal matrix protein abundance in Raw264.7 cells.** (**A**) RAW246.7 cells were treated with 5 mM, 10 mM, or 20 mM of 3-AT for 24 h and catalase activity was measured. (**B**) Cells were treated as in (**A**) and followed by immunoblotting with antibodies as shown. (**C**) Cells were treated as in (**A**), followed by subcellular fractionation, and by immunoblotting for Pex5, PMP70, β-actin and UBXD8. (**D**) Cells were treated as in (**A**) and mRNA expression levels of Catalase, PMP70, ACOX1 and DBP were measured. (**E**) Cells were treated as in (**A**) and immunostaining was performed with anti-catalase (green) and anti-PMP70 (red). (**F**) The ratio of catalase positive PMP70 puncta to the total PMP70 puncta per cell was determined from (**E**) was calculated by dividing total number of catalase puncta to total number of PMP70 puncta per cell. At least 30 cells were counted in each experiment. All error bars represent the mean ± S.D. (n = 3, independent experiments), *p < 0.05.

abundance by immunoblotting (Fig 1B). Interestingly, 3-AT did not affect the expression level of peroxisome membrane protein PMP70 as expected for pexophagic events based on our previous report [18]. Rather, it decreased the expression of peroxisome matrix proteins, including catalase, DBP and ACOX1, in a dose dependent manner. Peroxisome matrix proteins contain peroxisomal targeting sequence (PTS1) which is imported by Pex5 [21]. The protein expression of PEX5 was slightly decreased by 3-AT and remained sustained in a concentration dependent manner (Fig 1B). Moreover, protein expression level of PTS2 import receptor, Pex7, remained unaltered in 3-AT treated cells. Furthermore, Pex14 required for docking of Pex5 and Pex7 was also not altered by 3-AT. These data suggest that a decrease in peroxisome matrix proteins in the presence of 3-AT may not be associated with the import system of peroxisome matrix protein. Moreover, the protein expression levels of essential biogenesis factors, such as Pex1, Pex3, Pex16 and Pex19, were not altered by 3-AT, suggesting that a decreases in expression of peroxisomal matrix proteins was not related with peroxisome biogenesis issues. Importantly, 3-AT neither induced lipidation of LC3 (LC3II) nor decreased p62 protein level (Fig 1B), indicating that autophagic degradation of peroxisomes was not occurred.

Recycling of Pex5 from peroxisomes to the cytosol is known to be regulated by the action of exportomer complex consisted of Pex1, Pex6 and Pex26 after delivery of PTS1 containing proteins into the peroxisome matrix [19]. Defective exportomer function results in the accumulation of Pex5 on the peroxisomal membrane which further decreases matrix protein import as well as induction of pexophagy [19]. Therefore, we initially speculated that 3-AT might impair peroxisomal exportomer system, resulting in accumulation of import receptors on the peroxisome membrane which might cause the scarcity of these import receptors in the cytosol for import activities and promote the degradation of matrix proteins in the cytosol. However, Pex5 was found to be in the cytosolic fraction and not in the membrane fraction in both control and 3-AT treated cells (Fig 1C). Moreover, Pex1 which is important component of Pex5 exportomer complex was not affected by 3-AT (Fig 1B). Together, these data suggest that a decrease in peroxisome matrix proteins might not be associated with defective exportomer function. Also, a decrease in the protein level of catalase, DBP and ACOX1 by 3-AT was not associated with reduced mRNA expression (Fig 1D). We further confirmed that 3-AT decreased the catalase positive PMP70 puncta in a dose dependent manner (Fig 1E). In addition, 3-AT did not affect the protein levels of mitochondrial matrix protein SOD2 and mitochondrial membrane protein Tomm20 (Fig 1B). Taken together, these data confirm that 3-AT results in a selective degradation of peroxisome matrix proteins which is not associated with decreased mRNA expression of matrix proteins, pexophagy, and defective import/export of PTS receptors.

## N-acetylcysteine prevents the 3-AT mediated selective degradation of peroxisomal matrix proteins

Since 3-AT decreased the catalase activity, we questioned whether a selective degradation of peroxisome matrix proteins is associated with ROS generation. Intracellular ROS accumulation was analysed using DCFH-DA staining. Surprisingly, treatment with 3-AT did not induce significant increase in green fluorescence signal derived from DCFH-DA as compared to control cells. Whereas cells treated with $H_2O_2$, used as a positive control for ROS generation showed a huge increase in green fluorescence signal (Fig 2A and 2B), suggesting that 3-AT was not efficient to contribute to global ROS generation. To investigate whether 3-AT increases peroxisomal ROS, HyPer-SKL plasmid was transfected into RAW264.7 cells. HyPer-SKL is a fluorescent sensor widely used to detect peroxisome localized ROS [3, 18, 19, 22]. As shown in Fig 2C and 2D, green fluorescent signal could be observed in control cells transfected with

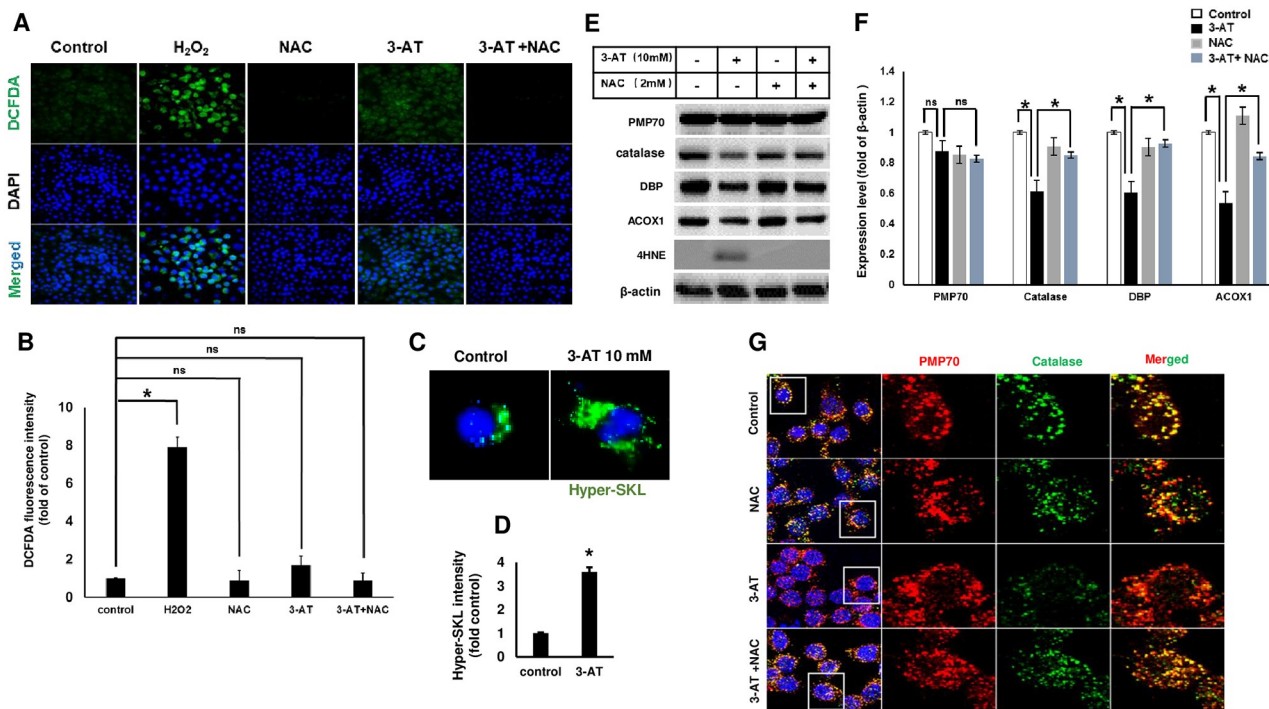

**Fig 2. Scavenging of peroxisomal ROS prevented the 3-AT mediated degradation of peroxisome matrix proteins. (A)** Cells treated with $H_2O_2$, 3AT (10 mM), NAC (2 mM) or co-treatment with NAC and 3-AT for 24 h. Cells were then treated with DCFH-DA (1 μM) and examined under fluorescence microscope. **(B)** Bar graph showing the fluorescent intensity of DCFH-DA from **(A)**. **(C)** Cells were transiently transfected with HyPer-SKL plasmid for 24 h and then treated with 3AT, for 24 h. **(D)** Peroxisomal ROS production was measured by fluorescence intensity of HyPer-SKL in the transfected cells from **(C)**. **(E)** Cells were treated with 3-AT alone, combination of 3-AT and NAC or NAC alone and followed by immunoblotting. **(F)** Densitometric analysis of band from **(E)**. **(G)** Cells were treated as in **(E)** and immunostaining was performed with anti-catalase and anti-PMP70. Representative images are shown. All error bars represent the mean ± S.D. (n = 3, independent experiments), *p < 0.05, ns: non significance.

HyPer-SKL. Treatment with 3-AT significantly increased fluorescent signal of HyPer-SKL in transfected cells. In addition, lipid peroxidation product 4-hydroxynonenal (4-HNE) was drastically induced by 3-AT whereas co-treatment of ROS scavenger, N-acetylcysteine (NAC), efficiently attenuated 3-AT mediated 4HNE production (Fig 2E). Also, cell viability was not affected by the tested concentrations of either NAC or co-treatment with 3-AT in MTT or TUNEL assay (S2A and S2B Fig). Furthermore, NAC efficiently restored the protein levels of peroxisome matrix proteins including catalase, DBP and ACOX1 (Fig 2E and 2F) as well as catalase positive PMP70 puncta (Fig 2G). Together, these data suggest that 3-AT induces the peroxisome ROS generation, which contributes to a selective degradation of peroxisomal matrix proteins.

## Catalase inhibition by 3-AT promotes leaky peroxisome and proteasomal degradation of peroxisome matrix proteins

Because 3-AT selectively promotes the degradation of peroxisome matrix proteins, we first speculated that possible cause behind this observation would be associated with re-translocation of protein from peroxisome matrix to the cytosol followed by proteasomal degradation as described for the quality control of mitochondrial matrix protein [23]. This event is likely to be associated with misfolded proteins and dysfunctional chaperones. Because 3-AT did not alter the abundance of PTS receptors which are also known to act as a chaperone [24], the

possibility of re-translocation of peroxisome matrix proteins into the cytosol followed by degradation was ignored. On the other hand, the leakage of peroxisome matrix proteins was previously reported in methylotrophic yeast *H. polymorpha* [25, 26]. Moreover, pore formation in the biological membrane has been linked to excessive lipid peroxidation, especially when membrane phospholipids are excessively oxidized [27]. Therefore, we also hypothesize that an increase in peroxisomal ROS generation by 3-AT may cause leaky peroxisome which then results in the leakage of matrix contents followed by proteasomal degradation. In order to test whether 3-AT promotes the proteasomal degradation of peroxisomal matrix proteins, proteasome inhibitor MG132 was co-treated with 3-AT in RAW264.7 cells. To exclude that 3-AT mediated degradation of peroxisomal matrix proteins was independent from pexophagy, chloroquine was also co-treated with 3-AT. As shown in Fig 3A, 3-AT mediated degradation of peroxisome matrix proteins, such as catalase, DBP and ACOX1, was markedly prevented by the addition of MG132 but not by chloroquine. These observations suggest that 3-AT promotes the proteasomal degradation of peroxisome matrix proteins rather than pexophagy. Regulation of catalase abundance by ubiquitin proteasome system has been previously reported [28]. Also, catalase is one of the matrix protein profoundly degraded by 3-AT. Therefore, the catalase ubiquitination assay was performed to further validate that 3-AT promotes the proteasomal of degradation peroxisome matrix proteins. RAW264.7 cells were transfected

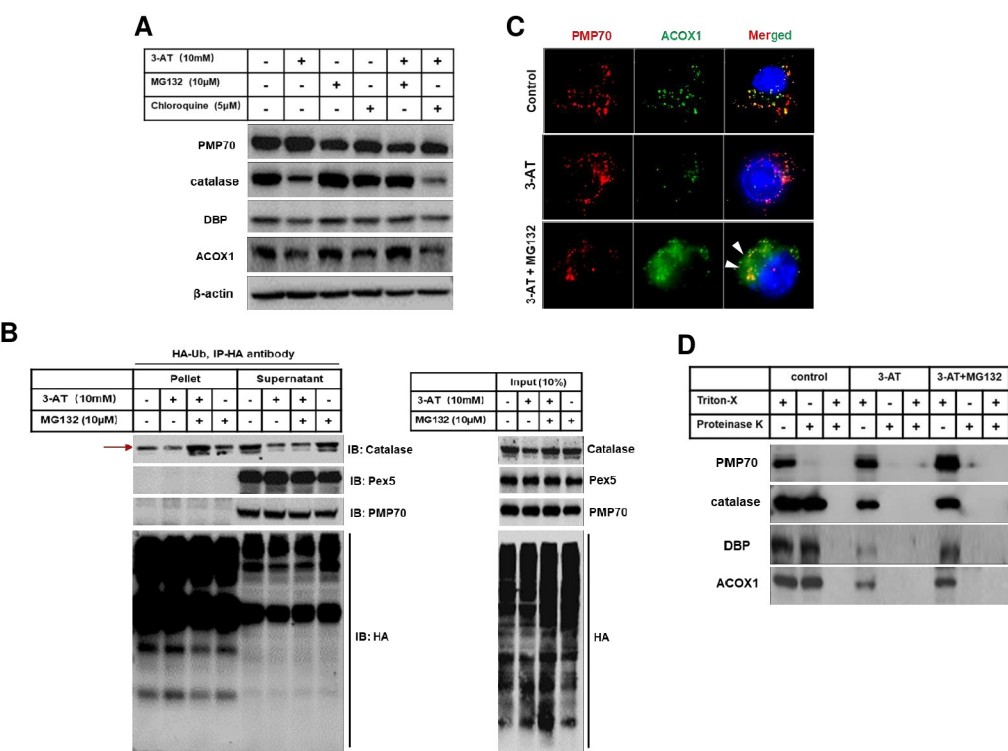

**Fig 3. 3-AT induced peroxisome leakage and promoted the ubiquitin mediated proteasomal degradation of peroxisomal matrix proteins. (A)** Cells were treated with 3-AT (10 mM), MG132 (10 μM), chloroquine (5 μM), combination of 3-AT and MG132, 3-AT or chloroquine for 24 h and followed by immunoblotting. **(B)** Cells were transfected with HA-Ub plasmid for 12 h. Cells were then treated with 3-AT, MG132 or combination of 3-AT and MG132 for additional 24 h. Cell lysates were then subjected for immunoprecipitation and immunoblotting. **(C)** Cells were treated with 3-AT alone, or combination of 3-AT and MG132 for additional 24 h and immunostaining was performed with anti-PMP70 (red) and anti-ACOX1 (green). **(D)** Cells were treated as in **(C)**, followed by Proteinase K protection assay, and immunoblotting for PMP70, Catalase, DBP and ACOX1.

with HA-Ub and followed by treatment with 3-AT alone, MG132 alone or both. Cell lysates were subjected to immunoprecipitation with anti-HA. As shown in Fig 3B, the pellet fraction represented all the proteins in the initial lysates bound to anti-HA whereas the supernatant represented proteins not bound with anti-HA. Ubiquitination of catalase was only induced by co-treatment with 3-AT and MG132 but not by MG132 alone. In contrast, catalase was mostly present in the supernatant in control and MG132 treated cells. These data suggest that 3-AT mediates catalase ubiquitination, which further promotes proteasomal degradation. Ubiquitination of Pex5 and PMP70 has been previously reported to be a crucial determinant of pexophagy [21, 29]. Ubiquitination of neither Pex5 nor PMP70 was not observed, further confirming that 3-AT mediated peroxisome matrix protein degradation is solely associated with ubiquitin proteasome system but not pexophagy. Smeared cytosolic ACOX1 staining was observed in cells co-treated with MG132 and 3-AT (Fig 3C), suggesting that 3-AT might promote leaky peroxisomes. In order to confirm that peroxisome matrix proteins are indeed degraded in the cytosol, proteinase K protection assay was carried out. As shown in Fig 3D, PMP70 was degraded by proteinase K without Triton X-100 whereas, peroxisome matrix proteins, including catalase, DBP and ACOX1, were only degraded by combination of proteinase K and Triton X-100 but not by proteinase K alone in control cells. These data suggest that peroxisome matrix proteins are primarily localized inside peroxisomes. In addition, their degradation by proteinase K required entry into peroxisomes through pores formed by Triton X-100 on the peroxisome membrane in control cells. Although, MG132 efficiently restored the abundance of peroxisome matrix proteins in the presence of 3-AT (Fig 3A), proteinase K was able to degrade them without the action of Triton X-100 (Fig 3D), suggesting that 3-AT promotes leakage of peroxisome matrix proteins into the cytosol followed by ubiquitin proteasomal degradation.

## Inhibition of catalase by 3-AT promotes the formation of IκBα-4HNE adduct

In this study, we found that 3-AT does not contribute much to the induction of global ROS production, rather it only induced peroxisomal ROS. We ask whether peroxisomal ROS modulates LPS induced inflammatory response in RAW264.7 cells. In this scenario it is of important to investigate whether LPS induced inflammation also contributes to the generation of ROS and how they are related with the expression of pro-inflammatory cytokines. Cells were treated with 100 ng/ml of LPS at different time interval, which revealed as no observable cytotoxicity (S3A and S3B Fig). Treatment with LPS induced mRNA expression of pro-inflammatory cytokines, including tumor necrosis factor-alpha (TNF-α), interleukin 1-beta (IL-1β) and IL-6, in a time dependent manner with maximum expression being observed at 12 h and subsided by 16 h (Fig 4A). In contrary, the protein expression of TNF-α and IL-6 was found to increase at 24 h of LPS treatment in the culture medium (Fig 4B). Next, we compared the time dependent effect of LPS and 3-AT on global ROS generation using DCFH-DA staining. Prior this experiment, cell viability was measured either by LPS alone or 3AT alone or combination of 3-AT and LPS by MTT assay or TUNEL staining, which did not showed any significant effects on cell death (S4A and S4B Fig). Although LPS treatment induced ROS generation at 24 h, ROS generation during 6 h and 12 h caused by LPS treatment was not evident. Whereas 3-AT did not contribute to ROS generation until 24 h in DCFH-DA staining (Fig 4C). These data suggest that LPS mediated ROS generation could be related with the increased abundance of pro-inflammatory cytokines in the medium with no relevance to mRNA expression of pro-inflammatory genes. Furthermore, 3-AT mediated peroxisomal ROS generation was evident as early as 6 h, increased in a time dependent manner, and persisted until 24 h as suggested by the increased fluorescent signal of HyPer-SKL (Fig 4D). Treatment with LPS did not alter

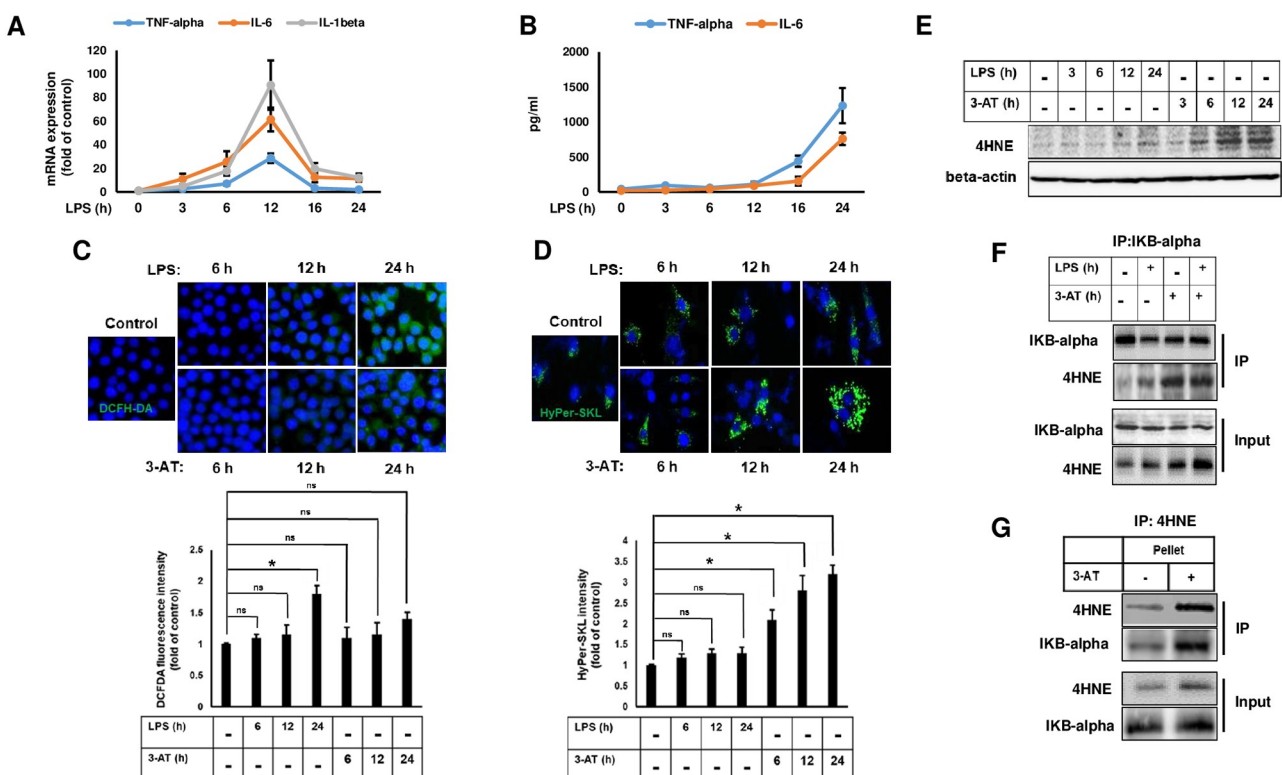

**Fig 4. 3-AT promotes the formation of 4HNE-IκBα adduct. (A)** Cells were treated with LPS (100 ng/ml) for different time points and mRNA expression levels of TNF-α, IL-1β, and IL-6 were measured. **(B)** Cells were treated with LPS for different time points and expression of TNF-α, IL-1β, and IL-6 was measured from the culture medium at each time points. **(C)** Cells were treated with LPS or 3-AT (10 mM) for different time points, added with DCFH-DA (1 μM), and examined under fluorescence microscope. Bar graph shows the fluorescent intensity of DCFH-DA. **(D)** Cells were transfected with HyPer-SKL and then treated with LPS or 3-AT for different time points. Peroxisomal ROS production was measured by fluorescence intensity of HyPer-SKL from the transfected cells. **(E)** Cells were treated with LPS or 3-AT for different time points and immunoblotting was performed with anti-4HNE. **(F)** Cells pre-treated with 3-AT for 12 h were further incubated with or without LPS for additional 12 h. Cells were then subjected to immunoprecipitation with anti-IκBα and followed by immunoblotting. **(G)** Cells treated with 3-AT for 12 h were subjected to immunoprecipitation with anti-4HNE and followed by immunoblotting. All error bars represent the mean ± S.D. (n = 3, independent experiments), *p < 0.05, ns: non-significance.

fluorescent signal of HyPer-SKL at any given time point, suggesting that LPS mediated ROS accumulation was not derived from peroxisomes. Together, these data suggest that LPS and 3-AT regulate ROS generation through different mechanisms.

Since we observed that 3-AT only induces peroxisomal ROS and 4HNE production without affecting DCFH-DA intensity, we speculate that lipid peroxidation could be the related with peroxisomal ROS and LPS induced ROS generation might not result in lipid peroxidation. As expected, 4HNE was not increased by LPS at any given time points (Fig 4E). In contrast, 3-AT induced 4HNE production as early as 6 h (Fig 4E), supporting our hypothesis that peroxisomal ROS induction is related with lipid peroxidation. Lipid peroxidation is a non-enzymatic reaction associated with oxidative stress during which lipids undergo oxidative damage by ROS producing free lipid radicals which maintain chain reaction for further peroxidation [30]. 4HNE is an one of the most abundant aldehyde product derived from oxidation of omega-6 polyunsaturated fatty acids such as arachidonic acids [28]. It has been reported that majority of ether phospholipids are enriched with arachidonic acids in macrophages [31, 32]. Furthermore, 4HNE has been known to form adduct with IκBα, which prevents its phosphorylation and directly interferes NF-κB activation [31]. Therefore, we propose that lipid peroxidation in RAW264.7 cells triggered by 3-AT might enhance the generation of 4HNE which then forms

4HNE-IκBα adduct. As expected, 3-AT drastically enhanced the interaction between 4HNE and IκBα in an immunoprecipitation reaction in the presence or absence of LPS (Fig 4F), implicating the formation of 4HNE- IκBα adduct. Further immunofluorescence staining clearly suggest the existence of interaction between 4HNE and IκBα in the presence of 3-AT (S5 Fig). In addition, immunoprecipitation by anti4HNE also detected IκBα in 3-AT treated cells (Fig 4G). Taken together, these data indicate that 3-AT promotes the formation of 4HNE-IκBα adduct.

### Inhibition of catalase by 3-AT prevents LPS induced NF-κB activation and mRNA expression of pro-inflammatory cytokines

Since phosphorylation of IκBα is required for nuclear translocation, we examined whether LPS induced IκBα phosphorylation was affected by 3-AT. As shown in Fig 5A, 3-AT signifi-cantly decreased IκBα phosphorylation, indicating that 3-AT directly inhibits LPS induced pro-inflammatory response by modulating IκBα phosphorylation. In addition, 3-AT also pre-vented the LPS mediated nuclear translocation of NF-κB (Fig 5B), suggesting that 3-AT inter-feres NF-κB activation and suppresses the production of pro-inflammatory cytokines at the transcriptional level. As expected, 3-AT significantly reduced the LPS mediated mRNA expres-sion of pro-inflammatory genes, including TNF-α, IL-1β and IL-6 (Fig 5C). Taken together, our data suggest that 3-AT prevents LPS induced pro-inflammatory response at the transcrip-tional level through inhibition of IκBα phosphorylation.

## Discussion

This study provides an interesting evidence that 3-AT mediated inhibition of catalase activity results in the accumulation of peroxisomal ROS which contribute to a selective degradation of

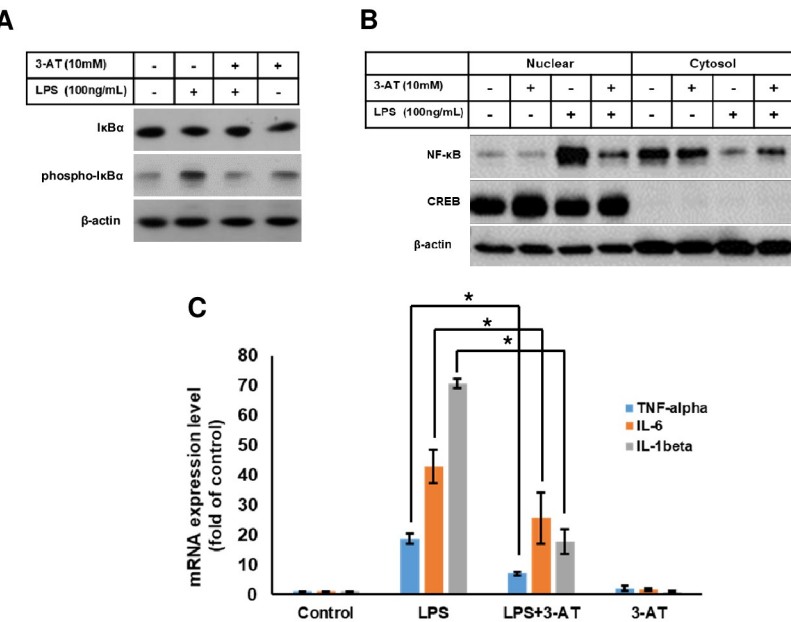

**Fig 5. 3-AT suppresses inflammatory response through promoting the formation of 4HNE-IκBα adduct. (A)** Cells pre-treated with 3-AT for 12 h were further incubated with or without LPS for additional 12 h. Cells were then subjected to immunoblotting with anti-IκBα and anti-phospho-IκBα. **(B)** Cells were treated as in **(A)**, followed by subcellular fractionation, and immunoblotting for NF-κB, β-actin and CREB. Cells were treated as in **(A)** and mRNA expression levels pro-inflammatory genes were measured by qPCR. All error bars represent the mean ± S.D. (n = 3, independent experiments), *p < 0.05.

peroxisome matrix proteins and suppression of LPS mediated pro-inflammatory response. A proposed mechanism for the interference of NF-κB activation presented here is strikingly different from our earlier reports [16, 17], in which we have suggested that 3-AT protects cells from cobalt chloride mediated cytotoxicity through preventing intracellular ROS generation in a catalase independent manner. Therefore, the observed discrepancies between our current study and earlier study could be attributed to various characteristics of different cell types. As mentioned earlier, ether phospholipids in macrophages are enriched with arachidonic acids which might play important role in making them vulnerable to lipid peroxidation than other cell types. Although induction of 4-HNE clearly suggests the extensive lipid peroxidation in macrophages, it is not yet clear how peroxisome derived ROS contribute to the formation of 4-HNE. It has been reported that leakage of $H_2O_2$ from peroxisomes could occur due to failure of peroxisomal antioxidant system [32–35]. Although it is plausible to question that ROS generated from peroxisomes might leak into the cytosol, which causes lipid peroxidation, the inefficiency of 3-AT for the production of global cellular ROS (Fig 2A) strongly suggests that the effect of 3-AT on lipid peroxidation is solely associated with peroxisomal dysfunction and probably associated with leaky peroxisomes. It is likely that leaky peroxisomes observed in the presence of 3-AT is directly related with impairment of NF-κB activation pathway. Here, we propose that disturbances in the antioxidant activities within peroxisomes result in lipid peroxidation of its' membrane components, which cause leaky peroxisomes leading to release matrix contents including matrix proteins and lipid peroxidation product 4-HNE into the cytosol. Although, mitochondria are an one of the most potent contributors of ROS to cells, 3-AT neither induces global ROS generation (Figs 2A and 4C) nor affects mitochondrial abundance (Fig 1B) and mitochondrial ROS (S6 Fig), which is consistent with our previous report that 3-AT itself did not induce ROS generation but it inhibits global ROS generation even in the presence of ROS inducers, such as cobalt chloride, in a catalase independent manner [16, 17]. In addition, 3-AT tends to inhibit LPS mediated transcriptional activation of pro-inflammatory cytokine genes by 12 h before the evidence for LPS mediated global ROS production at 24 h. Thus, it can be speculated that 4HNE derived from peroxisomes forms 4HNE-IκBα adduct which prevents IκBα phosphorylation and NF-κB activation. In the scenario, a selective degradation of peroxisome matrix proteins and suppression of NF-κB activation are not directly related rather both of them are the consequence of lipid peroxidation occurring at the peroxisomal membrane.

Our findings, together with earlier findings, demonstrate that peroxisomes mediated the regulation of NF-kB transcriptional activities may serve as a critical subcellular hub that promotes immune responses in animals [36].

Peroxisomes are equipped with several antioxidant enzymes besides catalase. It is surprising that inhibition of catalase is sufficient to elicit devastating effects on macrophage function. This notion suggests that macrophage peroxisomes are metabolically very active and might be continuously producing $H_2O_2$ which makes catalase an indispensable enzyme for the maintenance of macrophage function. The 3-AT mediated suppression of LPS induced pro-inflammatory response observed in the present study is associated with NF-κB activation but not with the failure to produce pro-inflammatory proteins derived by peroxisomal lipid or enhanced production of mediators for inflammation resolution.

## Supporting information

**S1 Fig. 3-AT induces cell death at different concentrations and time intervals.** Cell viability was determined in (A) RAW264.7 cells treated with 3-AT at concentrations indicated in for 24 h and (B) 10 mM of 3-AT at time points indicated by MTT assay. (C-D) TUNEL assay

obtained from RAW264.7 cells treated with 3-AT as in A and B. TUNEL-positive nuclei are indicated in green. Scale bar represents 50 μm.
(PDF)

**S2 Fig. Co-treatment with NAC and 3-AT does not induce cell death.** (A) Cell viability was measured by MTT assay in RAW264.7 cells treated with either NAC (2 mM) or co-treatment with NAC and 3-AT for 24 h. (B) TUNEL assay obtained from RAW264.7 cells treated A. Scale bar represents 50 μm.
(PDF)

**S3 Fig. LPS does not induce cell death at different time intervals.** (A) Cell viability was measured by MTT assay in RAW264.7 cells treated with 100 ng/ml of LPS at time points indicated. (B) TUNEL assay obtained from RAW264.7 cells treated A. Scale bar represents 50 μm.
(PDF)

**S4 Fig. Co-treatment with LPS and 3-AT does not induce cell death.** (A) Cell viability was measured by MTT assay in RAW264.7 cells treated with either 100ng/ml of LPS or 10 mM of 3-AT or both (LPS and 3AT) at time indicated. (B) TUNEL assay obtained from RAW264.7 cells treated A. Scale bar represents 50 μm.
(PDF)

**S5 Fig. 3-AT promotes the formation of 4HNE-IκBα adduct.** RAW 264.7 cells pre-treated with 10 mM of 3-AT for 12 h were further incubated with or without 100 ng/ml of LPS for additional 12 h. Cells were then subjected to immunofluorescence with anti-IκBα (green), anti-4-HNE (Red) and DAPI (blue). Arrow indicate the co-localization of IκBα with 4HNE (yellow).
(PDF)

**S6 Fig. Mitochondrial ROS was not induced in cells treated with 3-AT alone, LPS alone, or combination of two chemicals.** Representative red fluorescence image of MitoSOX Red from RAW 264.7 cells treated with 10 mM of 3-AT alone, 100 ng/ml of LPS alone, or combination of both for 24 h. Blue color represents DAPI stained. All images were processed and analyzed in a similar manner.
(PDF)

**S7 Fig. Uncropped western blot for Fig 1.**
(PDF)

**S8 Fig. Uncropped western blot for Fig 2.**
(PDF)

**S9 Fig. Uncropped western blot for Fig 3.**
(PDF)

**S10 Fig. Uncropped western blot for Fig 4.**
(PDF)

**S11 Fig. Uncropped western blot for Fig 5.**
(PDF)

## Acknowledgments

We thank Dr. Dong-Hyung Cho, Kyung Hee University, South Korea for providing us Hyper-SKL plasmid.

## Author Contributions

**Conceptualization:** Yizhu Mu, Yunash Maharjan, Raekil Park.

**Data curation:** Raghbendra Kumar Dutta, Xiaofan Wei, Jin Hwi Kim, Channy Park.

**Formal analysis:** Yizhu Mu, Yunash Maharjan, Raghbendra Kumar Dutta, Xiaofan Wei.

**Funding acquisition:** Raekil Park.

**Investigation:** Yizhu Mu, Yunash Maharjan.

**Methodology:** Yizhu Mu, Yunash Maharjan.

**Project administration:** Channy Park.

**Resources:** Jin Hwi Kim.

**Software:** Jinbae Son, Channy Park.

**Supervision:** Yunash Maharjan, Jin Hwi Kim, Jinbae Son, Raekil Park.

**Validation:** Yizhu Mu, Xiaofan Wei, Jinbae Son, Channy Park.

**Visualization:** Raghbendra Kumar Dutta, Raekil Park.

**Writing – original draft:** Yizhu Mu, Yunash Maharjan, Raekil Park.

**Writing – review & editing:** Raghbendra Kumar Dutta.

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
