## [Decision Letter · Decision Letter 0]

28 Oct 2020

PONE-D-20-29020

Pharmacological inhibition of catalase induces peroxisome leakage and suppression of LPS induced inflammatory response in Raw246.7 cell

PLOS ONE

Dear Dr. Park,

Thank you for submitting your manuscript to PLOS ONE. After careful consideration, we feel that it has merit but does not fully meet PLOS ONE’s publication criteria as it currently stands. Therefore, we invite you to submit a revised version of the manuscript that addresses the points raised during the review process.

The reviewers agree that your experiments are sound and support your conclusions, but they request additional information supported by additional experiments and improved statistical analysis. Reviewer 1 also requested that you edit your paper for improved clarity.

We look forward to receiving your revised manuscript.

Kind regards,

Alfred S Lewin, Ph.D.

Academic Editor

PLOS ONE

Journal Requirements:

4.PLOS ONE now requires that authors provide the original uncropped and unadjusted images underlying all blot or gel results reported in a submission’s figures or Supporting Information files. This policy and the journal’s other requirements for blot/gel reporting and figure preparation are described in detail at https://journals.plos.org/plosone/s/figures#loc-blot-and-gel-reporting-requirements and https://journals.plos.org/plosone/s/figures#loc-preparing-figures-from-image-files. When you submit your revised manuscript, please ensure that your figures adhere fully to these guidelines and provide the original underlying images for all blot or gel data reported in your submission. See the following link for instructions on providing the original image data: https://journals.plos.org/plosone/s/figures#loc-original-images-for-blots-and-gels.

Reviewers' comments:

Reviewer's Responses to Questions

**Comments to the Author**

1. Is the manuscript technically sound, and do the data support the conclusions?

Reviewer #1: Yes

Reviewer #2: Yes

2. Has the statistical analysis been performed appropriately and rigorously? 

Reviewer #1: Yes

Reviewer #2: No

3. Have the authors made all data underlying the findings in their manuscript fully available?

Reviewer #1: Yes

Reviewer #2: Yes

4. Is the manuscript presented in an intelligible fashion and written in standard English?

Reviewer #1: No

Reviewer #2: Yes

5. Review Comments to the Author

Reviewer #1: The article “Pharmacological inhibition of catalase induces peroxisome leakage and suppression of LPS induced inflammatory response in Raw246.7 cell” by Mu et al report the role of catalase and effects of peroxisome derived ROS on lipopolysaccharide (LPS) mediated inflammatory pathway are largely unknown. They showed that inhibition of catalase by 3-aminotriazole (3-AT) results in the generation of peroxisomal ROS, that leads to the leakage of peroxisomal matrix proteins into the cytoplasm in RAW246.7 cells. Furthermore, the author showed that 3-AT promotes the formation of 4HNE-IκBα adduct which directly interferes with the LPS induced NF-κB activation. The paper is interestingly and deserve publication alsthou few points should be addressed:

Major revision

1. The survival of the cells under each chemical treatment at each time point and under the selected concentrations should be shown (apoptosis assay or Cas3 staining)

2. The interaction between 4HNE and IκBα should be shown also by immunofluorescence

3. The methods and software used to quantify pictures and blots should be described in details in the methods

Minor revision:

1. The English overall the text needs improvement. Mainly in the abstract, introduction and discussion.

2. In the abstract define HNE

3. The reference Di Cara et al, Immunity 2017 should be cited in discussion since in this paper it was demonstrated for the first time the role of peroxisomes the role of peroxisomes redox and peroxisomal Catalase in mediating the regulation of NF-k��transcriptional activity in macrophages of Drosophila melanogaster.

Also the reference below is not correctly reported the first author is: Di Cara F Andreoletti P, Trompier D, Vejux A, Bülow MH, Sellin J, et al. Peroxisomes in

Immune Response and Inflammation. Int J Mol Sci. 2019;20(16):3877. pmid: 31398943

Cara FD, Andreoletti P, Trompier D, Vejux A, Bülow MH, Sellin J, et al. Peroxisomes in

Immune Response and Inflammation. Int J Mol Sci. 2019;20(16):3877. pmid: 31398943

Reviewer #2: This paper by Mu, Maharjan and Dutta et al., concerns the ability for catalase inhibition to suppress inflammatory responses in the macrophage cell line RAW 264.7. The authors found that 3-AT induced leakiness in peroxisomes, resulting in the production of peroxisomal ROS, 4HNE production and subsequent inhibition of NFkB signaling by adduct formation in the NFkB pathway. The evidence comes from conventional molecular and cell biological assays and is carefully performed, reported and interpreted. This study will be of interest to scientists studying reactive oxygen signaling, peroxisome biology and inflammation. I have some comments which the authors are encouraged to address:

• The authors suggest that lipid peroxidation is not dependent on processes that are induced by LPS treatments or the associated ROS. The authors are encouraged to discuss or provide additional experimental support regarding how their results in Figure 4 differ from previously published work by Yang et al., J Neuroinflammation 2018, which found that LPS did indeed induce 4-HNE production in microglial cells (doi: 10.1186/s12974-018-1232-3). See also, Choi et al J Neuroscience 2007 (doi: 10.1523/JNEUROSCI.5417-06.2007) and Chen et al Frontiers in Immunology 2019 (doi: 10.3389/fimmu.2019.01904) – this last one used bone marrow-derived macrophages and thus should be given particular note as it is unlikely simply because of the differences between macrophages and microglia (which are already considered quite similar in function).

• The authors focused only on whole-cell ROS production and peroxisome-specific ROS production, but did not give any attention to mitochondrial ROS. mROS is known to be significantly induced by LPS treatment, participates in lipid peroxidation, and has been shown to be induced by 3-AT treatment in vitro (Walton and Pizzitelli, Frontiers in Physiology 2012, doi: 10.3389/fphys.2012.00108). Mitochondria are one of the most potent contributors of ROS to the cell, and supporting evidence on the role of mitochondrial ROS in this process would provide significant strength to this study.

• Improvements must be made to the statistical analyses. The authors say in the methods that all statistics were performed by two-tailed T-tests, but throughout the manuscript are assessing more than two groups, which should be a one-way ANOVA (as in Figure 1A, D), or two variables (as in Figure 4C, D), which should be a two-way ANOVA, and in both cases statistics should be evaluated with an appropriate post-hoc analysis (Tukey, Holm-Sidak, or Fisher).

• Wherever single cell data is produced (eg., Figure 1F, Figure 2B, Figure 2D, etc) the authors should indicate how the fluorescence was quantified (using ImageJ? Some other software?) and more importantly how many cells were quantified by replicate and whether the bars and errors are indicative of the average of single cell data over numerous biological replicates.

Minor

• More information is needed regarding the Hyper-SKL and HA-ubiquitin plasmids. Source? Backbone? What were the transfection parameters?

• What is the specific name of the reverse transcription kit used?

• Please provide specific details on the protease and phosphatase inhibitors used.

• Although it is a common error, the authors are encouraged to use the correct name for the cell line, RAW 264.7. Repeatedly throughout the manuscript, and even in the title, the authors refer to RAW 246.7 cells. A cell line by this name does not exist. See: https://www.atcc.org/products/all/tib-71.aspx

• Some of the Western Blot images are extremely digitized/pixelated. Wherever possible, the authors are encouraged to include higher resolution images (eg, Supplemental Figure 2, the uncropped blots for Figure 2F – PMP70 and B-actin are inappropriately pixelated).

6. PLOS authors have the option to publish the peer review history of their article (what does this mean?). If published, this will include your full peer review and any attached files.

Reviewer #1: No

Reviewer #2: No

---

## [Author Response · Author response to Decision Letter 0]

4 Dec 2020

Dear Editor 

PLOS One

We thank the reviewers for the thorough assessment of the previous version of our manuscript. In response to the reviewers’ comments, we modified our manuscript and are modified text are highlighted in Yellow color of manuscript. We believe that this modified version of manuscript would effectively address the reviewers’ concerns. 

We have revised our manuscript according to the reviewers’ comments. Our point-by-point responses to reviewers’ comments and suggestions are provided below. We appreciate your time and consideration of this manuscript.

Sincerely

Raekil Park, M.D., Ph.D.

Laboratory of Peroxisomes & Lipid Metabolism, 

Department of Biomedical Science and Engineering, 

Gwangju Institute of Science and Technology, 

#309 Dasan Bldg., 123 Cheomdangwagi-ro, Gwangju 61005, Republic of Korea

Email: rkpark@gist.ac.kr

Phone: +82-62-715-5361, Fax: +82-62-715-5309

---

## [Decision Letter · Decision Letter 1]

16 Dec 2020

PONE-D-20-29020R1

Pharmacological inhibition of catalase induces peroxisome leakage and suppression of LPS induced inflammatory response in Raw 264.7 cell

PLOS ONE

Dear Dr. Park,

Thank you for submitting your manuscript to PLOS ONE. After careful consideration, we feel that it has merit but does not fully meet PLOS ONE’s publication criteria as it currently stands. Therefore, we invite you to submit a revised version of the manuscript that addresses the points raised during the review process.

Please demonstrate the interaction between 4HNE and IκBα  by Immunofluorescence as requested by reviewer 1.

We look forward to receiving your revised manuscript.

Kind regards,

Alfred S Lewin, Ph.D.

Academic Editor

PLOS ONE

Reviewers' comments:

Reviewer's Responses to Questions

**Comments to the Author**

1. If the authors have adequately addressed your comments raised in a previous round of review and you feel that this manuscript is now acceptable for publication, you may indicate that here to bypass the “Comments to the Author” section, enter your conflict of interest statement in the “Confidential to Editor” section, and submit your "Accept" recommendation.

Reviewer #1: (No Response)

Reviewer #2: (No Response)

2. Is the manuscript technically sound, and do the data support the conclusions?

Reviewer #1: Yes

Reviewer #2: Yes

3. Has the statistical analysis been performed appropriately and rigorously? 

Reviewer #1: Yes

Reviewer #2: Yes

4. Have the authors made all data underlying the findings in their manuscript fully available?

Reviewer #1: Yes

Reviewer #2: Yes

5. Is the manuscript presented in an intelligible fashion and written in standard English?

Reviewer #1: Yes

Reviewer #2: Yes

6. Review Comments to the Author

Reviewer #1: (No Response)

Reviewer #2: The authors have sufficiently addressed my concerns and I believe this manuscript is now suitable for publication. However, I would encourage the authors to include the MitoSOX experiment provided in the rebuttal statement as this is relevant data for readers of the manuscript and will contribute to a better understanding of the authors' science and to the field in general.

7. PLOS authors have the option to publish the peer review history of their article (what does this mean?). If published, this will include your full peer review and any attached files.

Reviewer #1: No

Reviewer #2: No

---

## [Author Response · Author response to Decision Letter 1]

4 Jan 2021

January 4, 2021

Dear Editor 

PLOS One

We thank the reviewers for the thorough assessment of our manuscript. In response to the editorial office, we removed funding related text from the manuscript. Although we would like to add Funding statement as:

This work was supported by 

• National Research Foundation of Korea (NRF) under grants funded by the Korean government No. 2018R1A5A1024340 and 2019R1A2C2086080 and 

• "GIST Research Institute (GRI) IIBR" grant funded by Gwangju Institute of science and technology (GIST) in 2020

We appreciate your time and consideration of this manuscript.

Sincerely

Raekil Park, M.D., Ph.D.

Laboratory of Peroxisomes & Lipid Metabolism, 

Department of Biomedical Science and Engineering, 

Gwangju Institute of Science and Technology, 

#309 Dasan Bldg., 123 Cheomdangwagi-ro, Gwangju 61005, Republic of Korea

Email: rkpark@gist.ac.kr

Phone: +82-62-715-5361, Fax: +82-62-715-5309

---

## [Editor Report · Decision Letter 2]

8 Jan 2021

Pharmacological inhibition of catalase induces peroxisome leakage and suppression of LPS induced inflammatory response in Raw 264.7 cell

PONE-D-20-29020R2

Dear Dr. Park,

We’re pleased to inform you that your manuscript has been judged scientifically suitable for publication and will be formally accepted for publication once it meets all outstanding technical requirements.

Kind regards,

Alfred S Lewin, Ph.D.

Section Editor

PLOS ONE
---

## [Editor Report · Acceptance letter]

15 Jan 2021

PONE-D-20-29020R2 

Pharmacological inhibition of catalase induces peroxisome leakage and suppression of LPS induced inflammatory response in Raw 264.7 cell 

Dear Dr. Park:

I'm pleased to inform you that your manuscript has been deemed suitable for publication in PLOS ONE. Congratulations! Your manuscript is now with our production department. 

Kind regards, 

on behalf of

Dr. Alfred S Lewin 

Section Editor

PLOS ONE